

# *Helicobacter pylori* prevalence in healthy Mexican children: comparison between two non-invasive methods

Verónica I. Martínez-Santos[1], Manuel Hernández Catalán[2],
Luis Octavio Ojeda Salazar[2], Octavio Andrei Orozco Gómez[2],
Sandra Ines Lorenzo[2], Rayver Santos Gómez[3],
Norma S. Romero-Castro[4], Roxana Reyes Ríos[5],
Dinorah Nashely Martinez Carrillo[2] and Gloria Fernández-Tilapa[2]

[1] Cátedras CONACyT-Universidad Autónoma de Guerrero, Chilpancingo, Guerrero, México
[2] Laboratorio de Investigación Clínica, Facultad de Ciencias Químico-Biológicas, Universidad Autónoma de Guerrero, Chilpancingo, Guerrero, Mexico
[3] Ih Max Gabinete de Diagnóstico, Universidad de Valle de Guerrero, Chilpancingo, Guerrero, Mexico
[4] Facultad de Odontología, Universidad Autónoma de Guerrero, Acapulco, Guerrero, Mexico
[5] Escuela Superior de Ciencias Naturales, Universidad Autónoma de Guerrero, Chilpancingo, Guerrero, Mexico

Corresponding author
Gloria Fernández-Tilapa,
gfernandezt@uagro.mx

## ABSTRACT

**Background:** *Helicobacter pylori* detection in asymptomatic children with suspected infection or with symptoms that suggest gastric pathology is problematic, since most of the methods depend on the endoscopic study, an invasive and expensive method. Non-invasive methods can be a feasible alternative but must be validated. The purpose of this study was to evaluate the concordance between *H. pylori* DNA detection in saliva and dental plaque by PCR, with antigen detection in stool by immunochromatography, among asymptomatic children in the state of Guerrero, Mexico.

**Methods:** Dental plaque, saliva, and stool samples were obtained from 171 children between 6 and 12 years old. *H. pylori* detection in saliva and dental plaque was performed by PCR using specific primers for the 16S rRNA gene, while the detection in stool samples was performed by immunochromatography using the CerTest kit.

**Results:** We found an overall *H. pylori* prevalence of 59.6% (102/171). Of the *H. pylori* positive children 18% (20/111) were positive in saliva samples, 28.1% (34/121) in dental plaque samples, and 50.4% (71/141) in stool samples. A higher prevalence was found in girls (64.7%, $p = 0.002$). Although some of the children declared some dyspeptic symptoms, these were no related to *H. pylori*. In conclusion, we found a high prevalence of *H. pylori* in asymptomatic children and the highest proportion was detected by stool antigen test, which was the most feasible method to detect *H. pylori* infection.

## INTRODUCTION

*Helicobacter pylori* is a spiral-shaped Gram-negative bacterium that colonizes the gastric mucosa in humans. A long-lasting infection with this bacterium can cause chronic gastritis, peptic ulcer, and gastric cancer (*Peek & Crabtree, 2006*). This bacterium can be acquired in childhood, usually before the first ten years of life, during which it can cause a transitory infection and be cleared spontaneously (*Malaty et al., 2002*; *Duque et al., 2012*). If not cleared, *H. pylori* can cause gastric disorders, like peptic ulceration, abdominal pain without peptic ulceration, gastroesophageal reflux disease, and recurrent abdominal pain; as well as extra-gastric issues, like low growth rate, iron deficiency, and iron deficiency anemia (*Campbell & Thomas, 2005*; *Rajindrajith, Devanarayana & de Silva, 2009*). However, an overwhelming majority of infected children are asymptomatic (*Jones et al., 2017*). Children can acquire this bacterium from person to person, through the oral-oral or fecal-oral route, although it could also be acquired through contaminated food and water, as well as infected animals (*Khalifa, Sharaf & Aziz, 2010*). The risk factors that influence the prevalence of *H. pylori* infection include low socioeconomic level, poor hygiene conditions, overcrowding, family history of parental gastric disease, and conditions that favor the person-to-person contact (*Hasosah et al., 2015*).

The frequency of *H. pylori* in Mexican children has decreased in recent years, while in 1998 the frequency of infection was found to be 50% (*Torres et al., 1998*), in 2012 it was 38% (*Duque et al., 2012*). However, the diagnosis of *H. pylori* infection in children is difficult to make, given that symptoms such as abdominal pain, nausea, vomiting, and occasionally diarrhea are unspecific (*Bosques-Padilla et al., 2018*).

Diagnostic tests to detect *H. pylori* are classified as invasive and non-invasive, depending on whether they require a gastric tissue sample obtained by endoscopy or not, respectively. Invasive tests include culture, PCR from biopsies, histopathology, and rapid urease test; while non-invasive tests include *H. pylori* antigens detection in the stool (stool antigen test, SAT), detection of antibodies anti-*H. pylori* in serum, PCR from oral samples, and urea breath test (*Guarner et al., 2010*). Of these, endoscopy and histological examination of gastric biopsies remain the gold standard for diagnosis (*Campbell & Thomas, 2005*). Since gastric malignancy is uncommon in children, the use of non-invasive tests to diagnose *H. pylori* gastric infection is highly preferred. Due to the potential oral transmission of *H. pylori* among children and adults, detection of the bacterium in oral cavity samples has been proposed as a diagnostic test, even though it is still not clear whether the oral cavity is another niche for the bacteria, or it is only an entry pathway (*Al Sayed et al., 2014*). The aim this work was to evaluate the concordance between *H. pylori* DNA detection in saliva and dental plaque samples by PCR, with antigen detection in stool by immunochromatography, among asymptomatic children in the state of Guerrero, Mexico.

## MATERIALS AND METHODS

### Population

One hundred and seventy-one children between 6 and 12 years old were invited to participate in the study. The parents were informed about the importance of early

detection of *H. pylori*, those who agreed to participate signed informed written consent and were asked for a stool sample of their child. In addition, a poll was applied in order to obtain information about the children's living conditions and oral hygiene. Teacher Nanci Aneli Cabañas Villanueva, principal of the elementary school "Esc. Prim. Daniel Delgadillo C.C.T 12DPR1692 J", allowed us to contact the students enrolled in the study and their parents. The project was approved by the Bioethics Committee of the Research Directorate of the Autonomous University of Guerrero (CB-004/2K20).

## Samples

Before obtaining dental plaque, 1–2 mL saliva samples were obtained by the passive drool method. The samples were collected in sterile propylene tubes containing extraction solution (Tris 10 mM pH 8, EDTA 20 mM pH 8, SDS 0.5%). Supragingival dental plaque samples were collected by a trained odontologist using sterile Gracey curettes (Hu-Friedy, Chicago, IL, USA), and placed in microcentrifuge tubes containing 750 μL of extraction solution. All samples were kept in ice while transported and then frozen at −20 °C until DNA extraction. Stool samples were collected by the parents in sterile containers. The samples were kept in ice until frozen.

## DNA extraction from dental plaque and saliva samples

Ten microliters of proteinase K (10 mg/mL) were added to the dental plaque and saliva samples previously dissolved in extraction solution. The samples were incubated at 65 °C for 12 h and the total DNA was obtained by the phenol:chloroform:isoamyl alcohol method. Then it was precipitated with ethanol and the pellet was resuspended in 30 μL of deionized water.

## *H. pylori* detection by PCR

*H. pylori* DNA in dental plaque and saliva samples was detected by standard PCR using primers HP16-219 (5′-GCTAAGAGATCAGCCTATGTCC-3′) and HPGR16SR (5′-CAATCAGCGTCAGTAATGTTC-3′) (*Chang et al., 2006*), which amplify a 522 bp fragment of the 16S rRNA gene. The reaction mix was performed in a 15 μL final volume, containing 2.5 mM $MgCl_2$, 0.25 mM dNTP's, 5 pmol of each primer, 1 U of Taq Platinum DNA polymerase (Invitrogen, Carlsbad, CA, USA), 1× buffer, and 150 ng of DNA. Amplification conditions were: 1 cycle at 94 °C for 5 min; 40 cycles at 94 °C for 30 s, 55 °C for 30 s, and 72 °C for 1 min; and 1 cycle at 72 °C for 7 min. PCR products were subjected to 2.5% agarose gel electrophoresis, stained with ethidium bromide, and visualized with ultraviolet light (UV). Reactions without DNA were included as negative controls and reactions with DNA from *H. pylori* strain 26695 were included as positive controls. DNA integrity was verified adding primers Sense (5′-CATTTGTCAGGTTCTTGATC-3′) and Antisense (5′-GAAGTTTAGTCTTCCCACTT-3′), which are specific for the promoter region of the human IL-1B gene and amplified a 305 bp fragment (*Roman-Roman et al., 2017*) (Fig. 1). Only IL-1B-positive samples were included in the study.

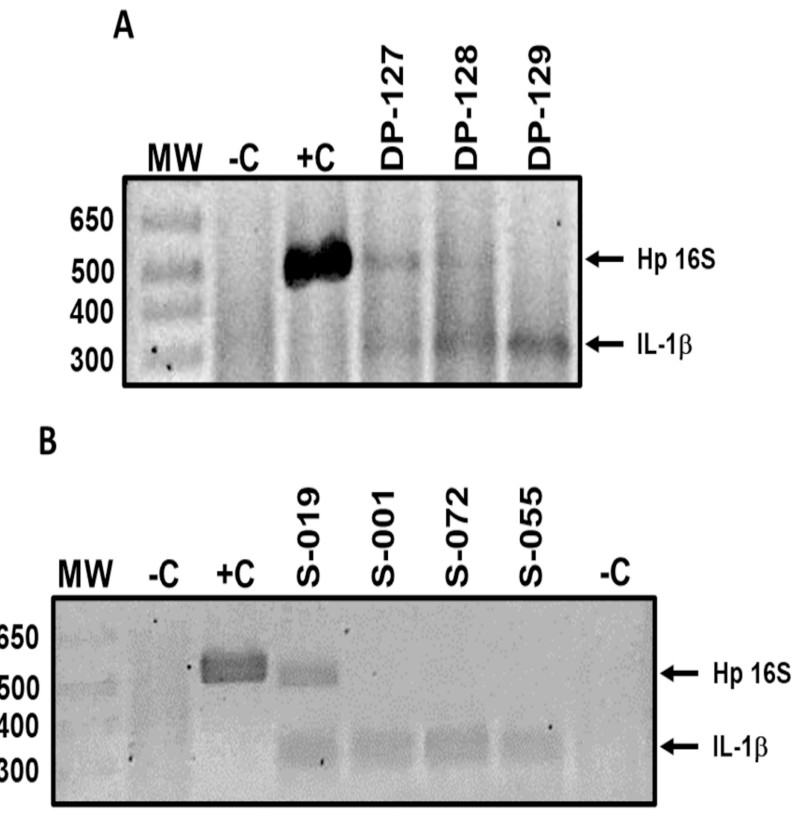

**Figure 1 Representative results of *H. pylori* detection in oral samples.** *H. pylori* DNA (Hp 16S) detection in (A) dental plaque and (B) saliva samples of asymptomatic children 6 to 12 years old. Lanes: MW-molecular weight marker; −C, negative control; +C, positive control; DP-127 and DP-128, *H. pylori* positive samples; DP-129, *H. pylori* negative sample; S-019, *H. pylori* positive sample; S-001, S-072, and S-055, *H. pylori* negative samples. IL-1β product was amplified as a DNA integrity control. DP, dental plaque; S, saliva.

### *H. pylori* antigens detection in feces

Detection of *H. pylori* antigens in stools samples was performed using the immunochromatographic one-step test CERTEST *H. pylori* (CerTest; Biotec, S.L., Zaragoza, España) following manufacturer's instructions. This test is a colored chromatographic immunoassay for the qualitative detection of *H. pylori* in stool samples. The strip consists of a nitrocellulose membrane pre-coated with mouse monoclonal antibodies against *H. pylori* on the test line, and with rabbit polyclonal antibodies against a specific protein, on the control line. The sample absorbent pad is sprayed with a solution of mouse monoclonal antibodies anti-*H. pylori* conjugated to red polystyrene latex, and a control solution of a specific binding protein conjugated to green polystyrene latex, forming colored conjugate complexes. A green line appearing in the control line in the results window is an internal control, which confirms sufficient specimen volume and correct procedural technique. The presence of this green line serves as (1) verification that sufficient volume is added, (2) that proper flow is obtained and (3) an internal control for the reagents.

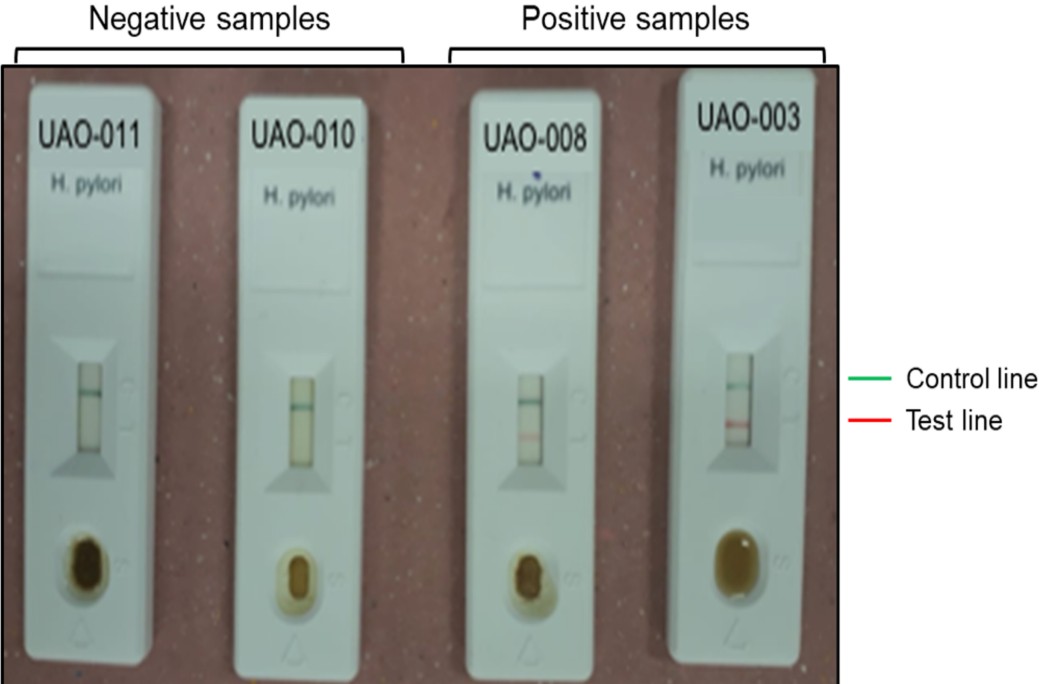

**Figure 2 Representative results of *H. pylori* detection in feces.** Negative (UAO-011 y UAO-010) and positive (UAO-008 y UAO-003) samples are shown. The green band is the control, and the red line indicates *H. pylori* presence.

The results were considered negative when only the green line appeared in the results window; positive when in addition to the green control line, a red line also appeared in the site marked with the letter *T* (test line); and invalid when the green line was completely absent, regardless of the appearance or not of the red line (Fig. 2).

## Statistical analysis

Statistical analysis of the data was performed using STATA v14.0 (StataCorp, College Station, TX, USA) software. Population characteristics, *H. pylori* frequency, and the relationship between *H. pylori* and gastric symptoms were compared using Fisher's exact test or the $X^2$ test when appropriate. Correlation between *H. pylori* detection methods was calculated using the Kappa index. Statistical significance was considered when $p < 0.05$.

## RESULTS

A total of 171 children, 45% (77/171) boys and 55% (94/171) girls, participated in the study, with an average age of 8.9 and 8.5 years old, respectively (age range between 6 and 12 years) (Table 1). Most of the children (91.8%, 157/171) were living in a rural community.

A child was considered to be *H. pylori*-positive when any of the samples gave positive results to bacterial DNA or antigens detection (either in one, two, or three samples). It is worth mentioning that some dental plaque and saliva samples were excluded because the sample or the purified DNA were insufficient, and not all the participants provided a stool sample. The overall prevalence of *H. pylori* was 59.6% (102/171) (Table 2).

**Table 1 Sociodemographic characteristics and *H. pylori* frequency in Mexican children aged 6 to 12 without gastric disease.**

| Characteristics | *H. pylori* negative *n* = 69 | *H. pylori* positive *n* = 102 | *p* value[Ω] |
|---|---|---|---|
| Sex, *n* (%) | | | **0.002** |
| Masculine | 41 (59.4) | 36 (35.3) | |
| Feminine | 28 (40.6) | 66 (64.7) | |
| Age, *n* (%) | | | 0.138 |
| 6–7 years | 12 (17.4) | 34 (33.3) | |
| 8–9 years | 34 (49.3) | 38 (37.3) | |
| 10–11 years | 17 (24.6) | 22 (21.6) | |
| 12 years | 6 (8.7) | 8 (7.8) | |
| Overcrowding, *n* (%) | | | 0.076 |
| No (≤2 people/room) | 27 (39.1) | 54 (52.9) | |
| Si (>2 people/room) | 42 (60.9) | 48 (47.1) | |

Note:
[Ω] $X^2$ test.
*p* values in bold are statistically significant.

**Table 2 *H. pylori* frequency in asymptomatic children by sample type.**

| *H. pylori* presence | Boys | Girls | Total | *p* value |
|---|---|---|---|---|
| Saliva, *n* (%) | | | | **0.002**[&] |
| Negative | 42 (95.4) | 49 (73.1) | 91 (82) | |
| Positive | 2 (5.6) | 18 (26.9) | 20 (18) | |
| Total | 44 (100) | 67 (100) | 111 (100) | |
| Dental plaque, *n* (%) | | | | **0.038**[Ω] |
| Negative | 41 (82) | 46 (64.8) | 87 (71.9) | |
| Positive | 9 (18) | 25 (35.2) | 34 (28.1) | |
| Total | 50 (100) | 71 (100) | 121 (100) | |
| Stool, *n* (%) | | | | 0.786[Ω] |
| Negative | 29 (48.3) | 41 (46.1) | 70 (49.6) | |
| Positive | 31 (51.7) | 40 (53.9) | 71 (50.4) | |
| Total | 60 (100) | 81 (100) | 141 (100) | |
| *H. pylori* status, *n* (%) | | | | **0.002**[Ω] |
| Negative | 41 (53.3) | 28 (29.8) | 69 (40.4) | |
| Positive | 36 (46.8) | 66 (70.2) | 102 (59.6) | |
| Total | 77 (100) | 94 (100) | 171 (100) | |

Notes:
[&] Fisher's exact test.
[Ω] $X^2$ test.
*p* values in bold are statistically significant.

Our results showed that the frequency of *H. pylori* was higher in girls, 38.6% (66/171) than in boys, 21% (36/171), and that there was no difference in prevalence among age groups (Table 1). Of the 102 *H. pylori*-positive cases, 64.7% (66) were girls. Overcrowding was not a risk factor for *H. pylori* infection in this population. An overcrowding condition was considered when three or more people slept in the same room.

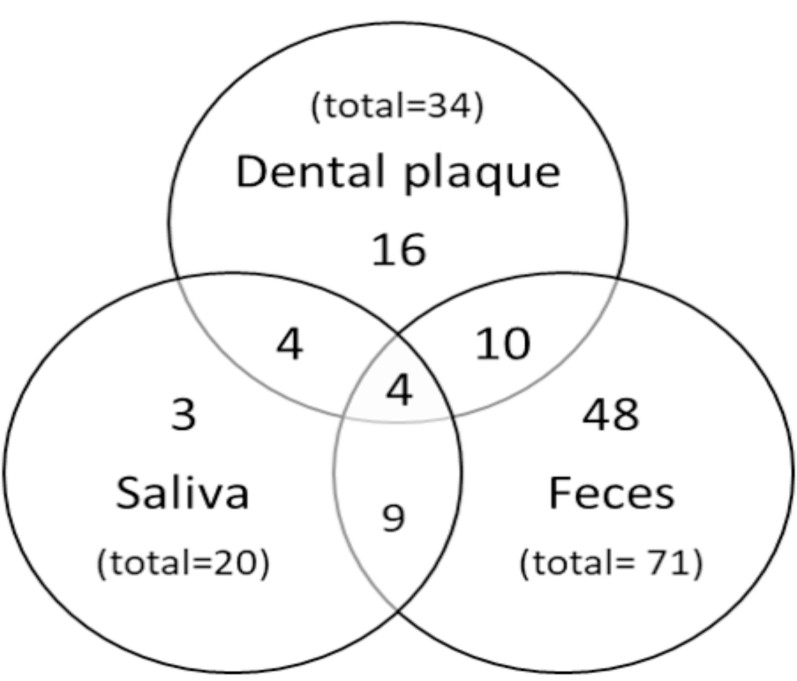

**Figure 3 Distribution of positive results.** Venn diagram showing the number of *H. pylori* positive children according to sample type.

### *H. pylori* frequency by sample type

In order to determine the frequency of *H. pylori*, 111 samples of saliva, 121 of dental plaque, and 141 of feces were analyzed. The highest *H. pylori* frequency was found in stool samples (50.4%) (Table 2). The prevalence of *H. pylori* in saliva and dental plaque was significantly higher in girls (26.9%, $p = 0.002$; 35.2%, $p = 0.038$, respectively) than in boys. In feces, no significant difference was found in *H. pylori* prevalence between both genders ($p = 0.786$). A variable percentage of children were *H. pylori*-positive in one, two, or three samples (Fig. 3). A total of 65.7% (67/102) of cases were positive in one sample, 22.5% (23/102) in two, and 3.9% (4/102) in all three samples.

### Concordance between detection methods

The prevalence of *H. pylori* infection detected by conventional PCR in dental plaque samples was lower than that obtained by SAT, and no agreement was found between the results ($κ = −0.1525$). A poor agreement was found when the results obtained by conventional PCR of saliva samples and SAT were compared ($κ = 0.09$) (Table 3).

### *H. pylori* and symptoms

All the children that participated in this study were considered healthy, since they reported no disease. Nevertheless, we asked them if they had any gastric symptom (Table 4). The percentage of *H. pylori*-positive children with frequent belching (16.7%) was significantly higher ($p = 0.035$) than that of the *H. pylori*-negative (5.8%). A total of 29.4% of the *H. pylori*-positive children referred suffering stomachache, however, there was no statistically significant difference with the *H. pylori*-negative children ($p = 0.952$).

**Table 3 Concordance among *H. pylori* DNA detection in saliva and dental plaque samples and SAT from children between 6 and 12 years old without gastric disease.**

| Dental plaque (*H. pylori* DNA) *n* (%) | Antigens in feces *n* (%) | | *p* value[Ω] |
|---|---|---|---|
| | **Negative** | **Positive** | |
| Negative | 27 (58.7) | 41 (74.5) | 0.091 |
| Positive | 19 (41.3) | 14 (25.5) | |
| Total | 46 (100) | 55 (100) | |
| Kappa index = −0.1525 | | | 0.9546 |
| Saliva (*H. pylori* DNA) *n* (%) | | | |
| Negative | 42 (85.7) | 42 (76.4) | 0.227 |
| Positive | 7 (14.3) | 13 (23.6) | |
| Total | 49 (100) | 55 (100) | |
| Kappa index = 0.09 | | | 0.1136 |

**Note:**
[Ω] $X^2$ test.

**Table 4 Relationship between *H. pylori* and gastritis symptoms.**

| Symptom | *H. pylori* negative | *H. pylori* positive | *p* value |
|---|---|---|---|
| Vomit, *n* (%) | | | 1.000[&] |
| No | 65 (94.2) | 97 (95.1) | |
| Yes | 4 (5.8) | 5 (4.9) | |
| Nausea, *n* (%) | | | 0.160[&] |
| No | 63 (91.3) | 99 (97.1) | |
| Yes | 6 (8.7) | 3 (2.9) | |
| Frequent belching, *n* (%) | | | **0.035**[&] |
| No | 65 (94.2) | 85 (83.3) | |
| Yes | 4 (5.8) | 17 (16.7) | |
| Stomachache, *n* (%) | | | 0.952[Ω] |
| No | 49 (71) | 72 (70.6) | |
| Yes | 20 (29) | 30 (29.4) | |
| Heartburn, *n* (%) | | | 0.742[&] |
| No | 66 (95.7) | 95 (93.1) | |
| Yes | 3 (4.3) | 7 (6.9) | |
| Diarrhea, *n* (%) | | | 0.205[&] |
| No | 63 (91.3) | 98 (96.1) | |
| Yes | 6 (8.7) | 4 (3.9) | |

**Notes:**
[&] Fisher's exact test.
[Ω] $X^2$ test.
*p* values in bold are statistically significant.

Between 2.9% and 16.7% of the *H. pylori*-positive children showed some symptom related to gastric infection by this bacterium, however, the percentages were not significantly different from those found in *H. pylori*-negative children. None of these symptoms were related to *H. pylori* infection.

## DISCUSSION

*H. pylori* infection occurs at early ages in developing countries and can cause several pathological conditions like gastritis, peptic ulcer disease, iron deficiency anemia, and growth faltering (*Rajindrajith, Devanarayana & de Silva, 2009*). Therefore, assessing children in regions with high prevalence of *H. pylori* infection is important to prevent related complications (*Ortiz-Princz et al., 2016*). Reference methods for *H. pylori* diagnosis are culture and histology, both invasive methods (*Talebi Bezmin Abadi, 2018*). However, in healthy or asymptomatic children, endoscopy is not justified, and so non-invasive tests are recommended. In this work we used three methods to determine *H. pylori* prevalence in apparently healthy Mexican children from the state of Guerrero. We found an overall frequency of 59.6%. This frequency is higher than those found in children from Mexico City and Monterrey (35–38%) in apparently healthy children (*Duque et al., 2012*; *Mendoza et al., 2014*; *Valdez-Gonzalez et al., 2014*; *Mendoza-Cantu et al., 2017*). This difference could be because the children that participated in this study are from a rural area, where the socioeconomic conditions are more precarious, besides the differences due to the geographical area and the methods used.

Contrary to previous reports that state that crowding is significantly associated with *H. pylori* infection, and that the frequency of infection increases with age (*Torres et al., 1998*; *Galal et al., 2019*), we found that *H. pylori* prevalence was very similar in all the age groups analyzed, and that crowding was not related to *H. pylori* frequency. On the other hand, gender was identified as a relevant characteristic for *H. pylori* acquisition, since most of the girls were *H. pylori* positive ($p = 0.002$). The reason for these findings is not known, and our results disagree with recent reports that have found no difference between genders, or male sex association with a higher *H. pylori* prevalence (*Duque et al., 2012*; *Valdez-Gonzalez et al., 2014*; *Ibrahim et al., 2017*; *Mendoza-Cantu et al., 2017*). The higher prevalence of *H. pylori* in girls (38.6%) than in boys (21%) is only in agreement with a previous report in Mexican population (*Torres et al., 1998*). This result was obtained by PCR of dental plaque (35.2%), followed by PCR of saliva samples (26.9%), while no difference between genders was found by SAT. It is proposed that the presence of *H. pylori* in the oral cavity is transitory and that it can be acquired by the oral-oral route. These results can be due to the fact that girls tend to be more affectionate and more likely to share food, candies, drinks, etc., and this contributes to a higher *H. pylori* presence in the oral cavity of girls than in boys.

In this work, *H. pylori* prevalence in dental plaque was 28.1%, which is lower than that found by *Valdez-Gonzalez et al. (2014)* and *Mendoza-Cantu et al. (2017)* (35% and 38%, respectively). In both studies, *H. pylori* detection was done by qPCR, while we used conventional PCR. The discrepancies can be attributed to differences in the sensitivity of the method used, in the origin of the population and in the age group studied. However, our results are relevant in the context of children's oral health. In Mexican children, the prevalence of gingivitis associated or not with dental plaque varies from 14% to 91.3% and varies between the children's geographical regions of origin and age (*Taboada Aranza & Talavera Peña, 2011*; *Rocha Navarro et al., 2014*). Although there are great

controversies about the relationship between *H. pylori* infection with gingivitis and periodontal disease, there are data that strengthen the hypothesis that the bacterium is associated with these pathologies (*Anand, Kamath & Anil, 2014*). In a study conducted by *Flores-Trevino et al. (2019)* in Mexican adults with periodontitis, 60.5% were *H. pylori*-positive. Together, these data suggest that 28.1% of asymptomatic children with *H. pylori* in their dental plaque, are at risk of gastric infection and/or of developing oral pathology associated with the bacteria.

The stool antigen test detects *H. pylori* antigens in the infected subject, which disappear once the bacteria are eradicated, so it is considered that this test provides evidence of an active infection (*Ricci, Holton & Vaira, 2007*). Our results suggest that there is some factor that favors *H. pylori* permanence on the oral cavity of girls or that the bacterium is more frequently acquired by them, but once it gets to the stomach it can effectively colonize the mucous of both sexes. Nevertheless, previous reports using qPCR (*Mendoza-Cantu et al., 2017*) or culture-qPCR (*Valdez-Gonzalez et al., 2014*; *Mendoza-Cantu et al., 2017*) to detect *H. pylori* DNA in dental plaque samples, found no difference in the prevalence of infection between girls and boys. More work is needed in order to verify the reasons for these differences, keeping in mind that the differences in frequency of infection could be due to the sample site, the sampling method, the *H. pylori* detection method, and the differences in the number of subjects included in the studies. Also, socioeconomic conditions, children's age, as well as the geographical region from which they come, are factors that can influence too.

Regarding the methods used, PCR of saliva and dental plaque have a sensitivity and specificity of 95% (*Bermúdez Díaz, Ernesto Torres Domínguez & Rodríguez González, 2009*), while the values for detection of antigens in feces are >94% and >99%, respectively (Certest *H. pylory Certest Biotec, S.L., 2019*). Despite having similar sensitivity values, a concordance analysis showed that there is poor to no agreement between the results from PCR and SAT. According to our results the antigen detection in feces is a better method for detection of *H. pylori* infected children (more sensitive), besides being faster and less expensive than molecular methods. In addition, the presence of *H. pylori* antigens in feces indicates an active gastrointestinal infection, while the presence of its DNA in oral cavity can be due to transient or dead bacteria. On the other hand, in children it is more feasible to obtain a stool sample than dental plaque and saliva samples with the required quality. However, a major disadvantage of SAT is that this method does not provide information about *vacA* genotype or *cagA* status of the bacterium, making it necessary to perform other methods.

*H. pylori* infection is mainly acquired during the first years of life, and the infected children show no symptoms (*Talebi Bezmin Abadi, 2018*). Even though the children that participated in this work were apparently healthy, we found that some of them had symptoms of gastritis, however none of these were related to *H. pylori*.

## CONCLUSIONS

In conclusion, we found a high *H. pylori* prevalence among asymptomatic children from 6 to 12 years of age from Guerrero, Mexico, and the SAT method was more sensible

to detect *H. pylori* infection than DNA detection from the bacterium in saliva or dental plaque.

## ACKNOWLEDGEMENTS

We would like to thank students from groups 501 and 502 (generation 2017–2022) of periodontics clinic class of the Faculty of Dentistry of the Autonomous University of Guerrero, for helping in taking dental plaque samples.

### Funding

This work was supported by the Program to Strengthen Educational Quality from the Secretary of Public Education of Mexico. The funders had no role in study design, data collection and analysis, decision to publish, or preparation of the manuscript.

### Grant Disclosures

The following grant information was disclosed by the authors:
Secretary of Public Education of Mexico.

### Competing Interests

The authors declare that they have no competing interests.

### Author Contributions

- Verónica I. Martínez-Santos conceived and designed the experiments, analyzed the data, authored or reviewed drafts of the paper, and approved the final draft.
- Manuel Hernández Catalán performed the experiments, analyzed the data, prepared figures and/or tables, and approved the final draft.
- Luis Octavio Ojeda Salazar performed the experiments, analyzed the data, prepared figures and/or tables, and approved the final draft.
- Octavio Andrei Orozco Gómez performed the experiments, analyzed the data, prepared figures and/or tables, and approved the final draft.
- Sandra Ines Lorenzo performed the experiments, prepared figures and/or tables, and approved the final draft.
- Rayver Santos Gómez performed the experiments, prepared figures and/or tables, and approved the final draft.
- Norma S. Romero-Castro conceived and designed the experiments, authored or reviewed drafts of the paper, and approved the final draft.
- Roxana Reyes Ríos conceived and designed the experiments, authored or reviewed drafts of the paper, and approved the final draft.
- Dinorah Nashely Martinez Carrillo analyzed the data, prepared figures and/or tables, authored or reviewed drafts of the paper, and approved the final draft.
- Gloria Fernández-Tilapa conceived and designed the experiments, analyzed the data, authored or reviewed drafts of the paper, and approved the final draft.

## Human Ethics

The following information was supplied relating to ethical approvals (i.e., approving body and any reference numbers):

The Bioethics Committee of the Research Directorate of the Autonomous University of Guerrero granted Ethical approval to carry out the study at the School of Chemical Biological Sciences (CB-004/2K20).

## Field Study Permissions

The following information was supplied relating to field study approvals (i.e., approving body and any reference numbers):

Teacher Nanci Aneli Cabañas Villanueva, principal of the elementary school "Esc. Prim. Daniel Delgadillo C.C.T 12DPR1692 J", allowed us to contact the students enrolled in the study and their parents.

## Data Availability

The raw data obtained from the PCR and stool antigen tests, as well as from the poll, are available in the Supplemental File.

## Supplemental Information

Supplemental information for this article can be found online at http://dx.doi.org/10.7717/peerj.11546#supplemental-information.

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
