# Peer review of "Helicobacter pylori prevalence in healthy Mexican children: comparison between two non-invasive methods"

_PeerJ, doi:10.7717/peerj.11546_

## Round 0.1 · original submission · Major Revisions

Please address critiques of the reviewers and revise the manuscript accordingly.

·

Basic reporting

The study objectives methodology is clear.

Experimental design

The experimental design also is good.

Validity of the findings

The authors found that the detection of H Pylori in the stool sample was more reliable compared to that in Saliva and dental plaque.

Additional comments

Well organised study.

·

Basic reporting

Abstract:
Although some of the children declared some dyspeptic symptoms, these were no (should be: not) related to H. pylori.

Main text:
Results:
Most of the children (91.8%, 157/171) live (should be: lived / were living) in a rural community.

Experimental design

No comment.

Validity of the findings

No comment.

Additional comments

No comment.

Reviewer 3 ·

Basic reporting

The paper entitled Helicobacter pylori prevalence in healthy Mexican children: comparison between two non-invasive methods conforms to professional quality standards in terms of its verbal expression.
This paper has sufficient bibliographic background although I would suggest another text on CagA and Vac Apara to make the bibliographic background information in Mexico more consistent, since there are few research groups in Mexico and this will help even more to strengthen the research groups.
The experiments show their methodology and results in a clear, consistent and easy to understand way for the reader, which coincide with the hypothesis test applied through statistical analysis.

Experimental design

The work is Original, and also meets the expectations and scope.
Besides considered as novel since in some way it complements much of the literature reported in the world and in addition to strengthening lines of research that support the knowledge here in Mexico and thus contributing to the strengthening of many other groups in the world.
In addition, it was observed that ethical care is taken since it shows the work record evaluated by an Ethics committee which, having been authorized by means of a support of CONACYT( Ministry of Investigation in Mexico) which should have also reviewed the berry nature of the document, especially for the reason of working with minors in Mexico which also shows the authorization of their parents or guardians

Validity of the findings

The statistical methods are adequate, clear and cosiso, also the conclusions are adequate since it mentions the limitations of the SAT method on the genotification of Cag A and VacA which is important to mention

Additional comments

Include an article on CagA and VacA also carried out with the patient population in Chiapas in Mexico since the future comparison of similar communities in the same country is interesting, which would provide future information of scientific interest.

Molecular detection of Helicobacter pylori based on the presence of cagA and vacA virulence genes in dental plaque from patients with periodontitis.
Flores-Treviño CE, Urrutia-Baca VH, Gómez-Flores R, De La Garza-Ramos MA, Sánchez-Chaparro MM, Garza-Elizondo MA. J Dent Sci. 2019 Jun;14(2):163-170. doi: 10.1016/j.jds.2019.01.010. Epub 2019 Mar 27.

Reviewer 4 ·

Basic reporting

The manuscript meets the basic criteria of publication, but there are some control experiments missing in its current version.

Experimental design

The design of the experiments employed in this work is sound, but negative control/loading control/normalization control need to be included.

Validity of the findings

The findings are relevant and might help to get new insights into the H pylori detection methods.

Additional comments

The manuscript entitled “Helicobacter pylori prevalence in healthy Mexican children: comparison between two non-invasive methods” presents a comparative study on three detection methods for H. pylori. Authors analyzed saliva and dental plaque using PCR method and stool samples by immunochromatography. Authors report that there is a high prevalence of H. pylori in asymptomatic children and amongst the multiple methods, detection by stool antigen test is most feasible method. The manuscript presents useful data here, but has some limitations. As the paper compares three different methods, my main concerns are pertaining to the technical details of the methods. Here are my comments:
1) Authors need to include information on the negative controls in the various methods tested. It is not clear to me if they were included. Authors need to specify this. If they were not included in the study, it would be important to include these. Also, did they have any normalization control/loading control? How did they rule out that the difference in the signal is not because of the difference in the input (bacterial load)? Please address these issues with respect to all the three methods described in the paper.
2) The methods are qualitative and results may be interpreted empirically. It would be impractical to represent these data in categorical (yes/no) format.
3) Authors may need to include a representative image (may be as supplementary data) for the PCR based detection method (both negative and positive samples).
4) Authors need to include additional data on the sensitivity and specificity of all the methods used. Did authors ensure the assays did not give any false positives and false negatives?
5) Author may consider presenting the data by Venn diagrams with overlaps representing the samples detected positive by multiple methods.
6) Did authors have technical replicates (all assays) included for each sample tested? It would be relevant to include a statement on this.

---

## Round 0.2 · accepted · Accept

In my view, the manuscript is amended in line with the reviewers' comments.

Reviewer 3 ·

Basic reporting

The English must use Clar, good standards and expression

Experimental design

It is very clear, simple and specific
It is stated how research fills an identified knowledge gap.

Validity of the findings

The information shown in this work is important and must be published since its scientific value will have a reflection or impact on the epidemiology and control of transmission mechanisms.

Additional comments

It is very important that this work be published since the information provided by this work is of great scientific value

Reviewer 4 ·

Basic reporting

The revised manuscript meets the basic criteria of publication. The work is well executed and technically sound. The data support their conclusions.

Experimental design

Experimental design is sound.

Validity of the findings

All the conclusions that are justified and based on clear results.

Additional comments

The revised manuscript addresses my concerns satisfactorily. I recommend the paper for acceptance in PeerJ.